# Spatio-Temporal Variation of Elemental Contamination and Health of *Mya arenaria* Clam in the Saguenay–St. Lawrence Marine Park

François Gagné [1,*], Chantale André [1], Samuel Turgeon [2] and Nadia Ménard [2]

[1] Environment and Climate Change Canada, 105 McGill, Montréal, QC H2Y 2E7, Canada; chantale.andre@ec.gc.ca

[2] Parks Canada, Saguenay–St. Lawrence Marine Park, 182, Rue de l'Église, Tadoussac, QC G0T 2A0, Canada; samuel.turgeon@canada.ca (S.T.); nadia.menard@pc.gc.ca (N.M.)

\* Correspondence: francois.gagne@ec.gc.ca

**Abstract:** The impacts of pollution and long-term effects of local clam populations are misunderstood in estuaries. The purpose of this study was to follow inorganic contamination in tissues, changes of physiological health indicators, such as condition factor (CF), growth index (GI), resistance in air emersion and dehydration rate, for 5 years in *Mya arenaria* clams. The sampling scheme comprised one reference site, two sites impacted by human activity (thereafter polluted) and one site recognized as a Saint-Lawrence Estuary (SLE) beluga whale feeding area without known pollution source (Baie Sainte-Marguerite (BSM)). This study revealed that the elemental contamination profiles in clams were increased but differed between the polluted and BSM compared to the reference site. At polluted sites, clams were contaminated by Ag (2.4-fold of reference site), Mn (2.5-fold) and V (6.3-fold). With respect to BSM, clams were mainly contaminated by Ce (2.5-fold), Co (2-fold), Ga (2-fold), La (2.8-fold), Hg (2.5-fold), Ni (2.2-fold), Sm (2-fold) and V (20-fold). This contamination profile suggests sources of pollution from particulate combustion products of gasoline/diesel, crude oil and urban inputs of pollution. The CF, GI and air survival time were all reduced in clams at the polluted sites, while only the CF and dehydration rates were decreased and increased, respectively, at BSM. Long-term analysis revealed that CF and GI tended to decrease over time with episodes of strong amplitude changes and became more resilient to air survival time. In conclusion, the long-term contamination of clams towards metals and elements could compromise the health status of local clam populations. The increased contamination of clams at BSM could represent a risk to the endangered SLE beluga whale population.

**Keywords:** *Mya arenaria*; condition factor; growth index; air survival; metals; rare earths



## 1. Introduction

The Saguenay–St. Lawrence Marine Park (Quebec, Canada), located at the confluence of the Saguenay Fjord and the SLE, is located downstream of many important urban and industrialized sites. The Marine Park is also crossed by the St. Lawrence Seaway and is the home of many commercial and tourism boating activities, namely, an important whale-watching industry [1]. Moreover, it supports recreational boat traffic, local harbors, and numerous industrial and municipal wastewater discharges. The continuous occurrence of environmental pollutants such as polyaromatic hydrocarbons, organotins, heavy metals and episodes of red tides (algal blooms) from increased nutrients in the SLE has been reported over the last decades [2–4]. *Mya arenaria* clams are commonly found in Saguenay Fjord and SLE mudflats and are well known to accumulate numerous contaminants, such as organotin [5], and many other compounds, such as arsenic and polyaromatic hydrocarbons [6]. In addition, clams are relatively long lived (12–15 y), sessile and serve as a food source to many organisms, such as rag worm *Neiris diversticolor* [7], gammarids, fish, birds

and even beluga whales in this area. This makes them ideal bio-sentinels to monitor the water and sediment quality of coastal ecosystems and marine-protected area health.

The ecotoxicological impact of local sources of pollution (municipal effluents, harbors, boat traffic and red tides) on *Mya arenaria* clams has been studied for many years [8,9]. These studies revealed that clam health was locally compromised at many levels, including neuroendocrine, DNA damage, immunocompetence and increased energy expenses, supporting the hypothesis on the negative impacts of marine and urban pollution on intertidal clams. A recent review investigated the use of simple and readily accessible health indices based on morphological parameters such as shell length, age and mass to determine the negative impacts of pollution on sediment quality [10]. These three simple physiological biomarkers were therefore proposed that were previously proved to underly many biochemical biomarkers, such as genotoxicity, oxidative stress, reproduction and immunocompetence. Indeed, biochemical/molecular biomarkers are undeniably key tools to understand the toxicity of pollution at the sub-cellular level, but their application in unspecialized laboratories is often constrained by budgetary limitations and a lack of expert personnel. This is especially true for more remote laboratories. These are the condition factor (CF: clam weight-g/shell length-mm), growth index (GI: shell length/age) and air survival time (days). The last endpoint was also called the "stress on stress" (SOS) response based on the assumption that bivalves undergoing stress by disease, contamination, anoxia or diet modifications, etc., are less able to withstand prolonged periods in air [11]. The evaluation of weight loss principally from dehydration was sometimes measured at the same time, providing another measure of this metric. This physiological biomarker has gained more interest in marine and even in freshwater bivalves in the context of water-level extremes (longer episodes of drought punctuated by intense precipitations) and more extreme temperature changes from climate change [12]. These three physiological biomarkers of diagnostic health status are straightforward to perform, requiring only an analytical balance and a ruler to measure mass and length. Since the long-term (multi-year) effects of pollution are less understood on feral *Mya arenaria* clam populations, the above approach appeared as an attractive tool to examine clam health in the context of routine surveys by remote laboratories. The present study constitutes, therefore, a direct application of this approach in clam populations under pollution by anthropogenic activities (marina, harbors), as confirmed by elemental contamination in tissues.

The purpose of this study was to examine the long-term changes in clam health status based on the above physiological biomarkers for 5 years. Elemental (including metals) contamination of clams were also analyzed in tissues in an attempt to identify key contamination signatures in clams collected at two impacted sites and an SLE beluga residency area relative to a reference site under no direct source of pollution (Figure 1). Since beluga whales feed on *Mya arenaria* clams, information on the contamination and health status of clams from a feeding area is needed to ensure the protection of this endangered species. This study also aimed to quantify the natural variation (5 and 95th percentiles) in CF, GI and air survival (SOS response) for over 5 years of data from the reference site in order to identify responses of clams at polluted sites and determine whether the BSM area is affected by pollution.

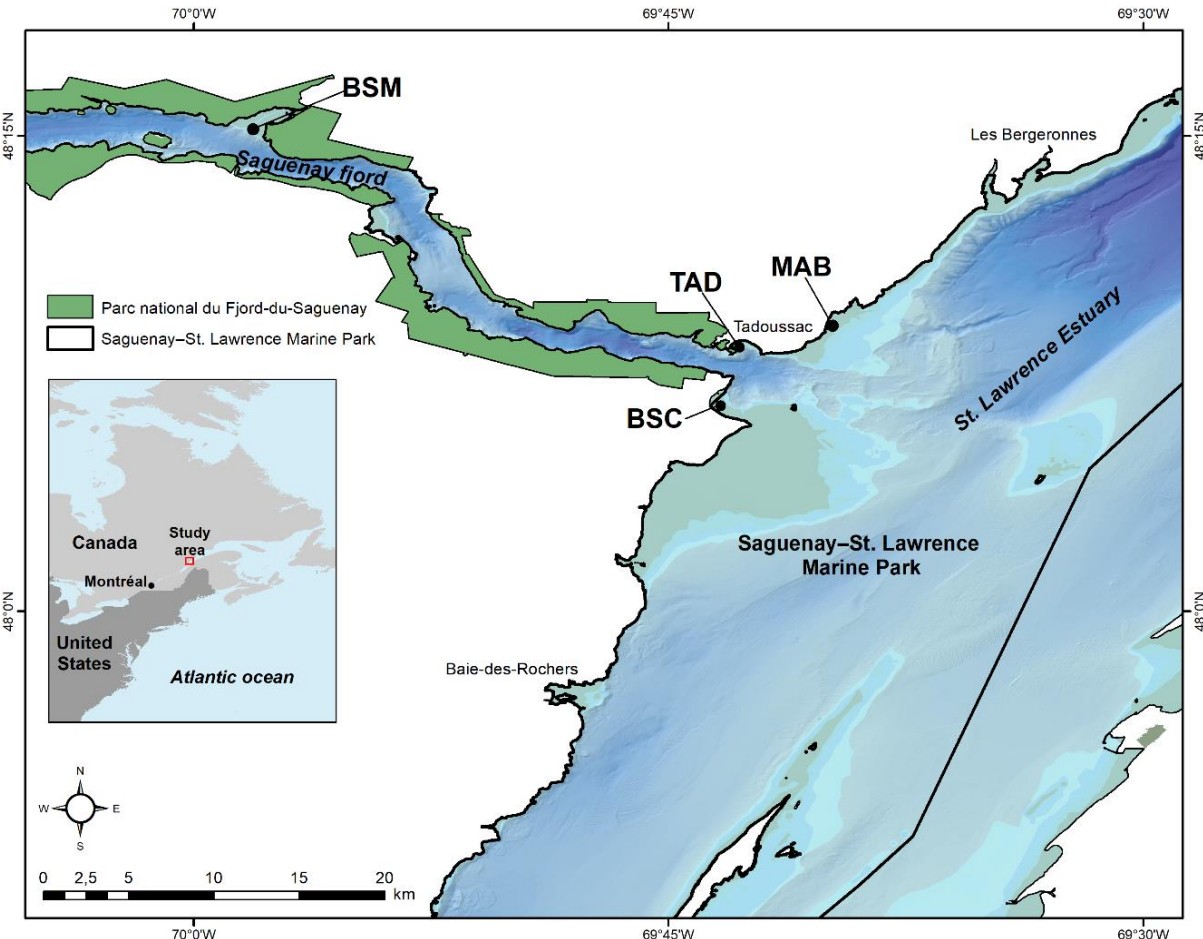

**Figure 1.** Study area. The location of four *Mya arenaria* intertidal sampling sites: TAD (Baie de Tadoussac), a site subject to wharf, marina and wastewater discharge pollution; MAB (Baie du Moulin-à-Baude) a reference site with no known or apparent source of pollution; BSC (Baie Sainte-Catherine), a site also subject to wastewater discharge and wharf pollution; BSM (Baie Sainte-Marguerite), an SLE beluga whale high residency area.

## 2. Methods

### 2.1. Clam Collection and Handling

Clams were collected at four different sites in the Saguenay–St. Lawrence Marine Park centered at the mouth of the Saguenay fjord area, as depicted in Figure 1. The sites TAD and BSC are considered polluted, as they support intense boat traffic and marina and wharfs for recreational and commercial purposes (e.g., whale-watching activities), while the site MAB is considered the reference site since it is under no direct source of pollution and anthropogenic activities. The site BSM is located 25 km upstream of Tadoussac in the Saguenay Fjord and is surrounded by the Parc national du Fjord-du-Saguenay (SÉPAQ) (Figure 1). BSM is well known for intensive summer use by SLE beluga [13], especially females and young [14], and it is considered relatively pristine (no maritime infrastructure) although it is under recreational and commercial boat traffic into the Saguenay Fjord. Clams (N = 40) over 50 mm were yearly collected during low tide during the last part of September from 2016 to 2020. Then, 15 of these 40 clams were randomly selected for morphological assessment (condition factor-CF and growth index-GI as described below). Clams were kept in sea water and refrigerated until analysis. Wet sediment was collected during low tide and sediment water was prepared by mixing 10 g of sediment with 10 mL of MilliQ water followed by centrifugation at $1000 \times g$ for 5 min. The following parameters were determined: granularity (sandy/gravel composition), surface water and sediment pH, salinity and dissolved oxygen.

### 2.2. Tissue Metal Content Analysis

For elemental analysis of clam tissues, the tissues from 6 individual clams were taken at each site and weight. The elemental analyses of tissues were performed at the National Laboratory of Environmental Testing (NLET) of Environment and Climate Change Canada [15] on the 2016–2018 samples. Briefly, 0.5 g soft tissues (wet) was placed in a 15 mL Quartz digestion vessel containing 4 mL of nitric acid. The vessel was placed in a high-pressure microwave oven for thermal digestion at 240 °C for 15 min. After cooling at room temperature, it was transferred to a 50 mL digestion tube and brought to 50 mL volume with a 1% hydrochloric acid blank solution. The digested sample was then analyzed by Inductively Coupled Plasma Mass Spectrometry (ICP-MS) for the following 45 elements using the main isotope: aluminum, antimony, arsenic, barium, beryllium, bismuth, bore, cadmium, calcium, cerium, cesium, chrome, cobalt, copper, europium, gadolinium, magnesium, manganese, mercury, molybdenum, neodymium, nickel, niobium, palladium, phosphorus, platinum, potassium, praseodymium, rubidium, ruthenium, samarium, scandium, selenium, silver, strontium, tellurium, terbium, thallium, tin, titanium, tungsten, uranium, vanadium, zinc, and zirconium. Data are expressed as µg/g wet weight.

### 2.3. Morphological Biomarkers and Air Survival Test

A total of 15 clams were analyzed for age, shell length, mass and air survival time as previously described [9]. The age was determined by counting the number of major grooves or annual rings on shells. The condition factor (CF) was calculated by the clam wet weight (g)/longitudinal shell length (mm), and the growth index was calculated by the shell length/age. For air survival times, 15 clams from each site were aged and weighted and their shell lengths were determined. The clams were then numbered with a permanent marker and placed in open plastic containers (one for each site) at 20 °C lined with humidified (sea water) paper towels at the bottom. Mussels were weighted each day to determine the weight loss (WL) related mostly to water loss, i.e., WL was considered to be dehydration rate. Clam death was determined by the capacity of closing their shells after manipulation. The lethal time (LT) was expressed in days required for death. The WL was calculated by the % of weight loss compared to day 0 normalized for clam mass: [(g at day 0- g at day $\times$)/g at day 0] $\times$ 100.

### 2.4. Data Analysis

A total of 15 clams were selected for morphological parameters (shell length, mass, age), air survival test and tissue metal analysis. The data were analyzed for normality and homogeneity of variance using the Shapiro–Wilks and Levene's tests. The sites were organized into three groups of sites: reference site (MAB), polluted sites (BSC and TAD) and the SLE beluga area of high residency (BSM). An analysis of variance (ANOVA) was performed on either sites or a group of sites followed by the Tukey's multiple comparison tests. In the case that the data were not normally distributed, a Kruskall–Wallis ANOVA followed by the Conovan-Iman's test as the multiple comparison tests were then performed. The data was expressed with the mean and 5–95th percentiles (5 years combined) to fix the normal range of the data spread at each site or group of sites. Correlation analysis was also performed using the Pearson-moment procedure. Time series analysis was achieved by Fourier transformation and rescaled range analysis (H exponent) to determine the long-term trends and seek out periodic changes in the individual responses (CF, GI and LT) for over 5 years. Significance was set at $p < 0.05$ and the SYSTAT software package was used (version 13, San Jose, CA, USA).

## 3. Results

A five-year survey of intertidal *Mya arenaria* clams was undertaken to follow changes in clam health status based on morphological parameters such as condition factor (clam mass/longitudinal shell length), age and growth index (longitudinal shell length/age). Moreover, the air survival time and dehydration rate were also analyzed, in keeping with

the study by André et al. [10]. First, the sites were characterized by measuring water pH, salinity, dissolved oxygen contents and granularity (results not shown). Over the 5-year sampling period, the water pH and oxygen saturation levels varied from 7.8 to 8.3 and 95–99%, respectively, regardless of the sites. Water salinity ranged between 28 and 34 g/L at BAU, BSC and TAD sites located in the SLE but dropped to between 21 and 24 g/L at the BSM site, which was located downstream of the Saguenay Fjord (Figure 1). The sediment mud flat was composed of sandy material containing 10–30% gravel (0.1–10 cm size) at the SLE sites but was mostly sand at the BSM site. The tissue-elemental loadings were also determined in clam tissues over the first 3 years of the survey to determine potential contamination problems (Table 1). Elemental analysis revealed that 30 elements revealed changes between the reference site with either the polluted or BSM sites. With respect to the polluted sites, the following elements were significantly increased (>2-fold are in **bold**): **Ag (2.4-fold)**, Ca (1.16-fold), Cd (1.2-fold), Cr (1.6-fold), **Mn (2.5-fold)**, Mo (1.6-fold), Nb (1.6-fold), Nd (1.7-fold), K (0.9-fold), Sm (1.2-fold) and **V (6.3-fold)**. The following elements were found at higher levels at the BSM compared to reference sites: Al (1.7-fold of reference site), Ba (1.7-fold), Be (1.4-fold), Ce (2.5-fold), Co (2-fold), Cu (1.3-fold), **Ga (2-fold)**, Gd (1.8-fold), Fe (1.7-fold), **La (2.75-fold)**, Pb (1.7-fold), Mg (0.7-fold), **Hg (2.5-fold)**, **Ni (2.2-fold)**, **Sm (2-fold)**, Sn (1.7-fold), U (1.3-fold), **V (20-fold)**, Zn (1.9-fold) and Zr (1.3-fold). The following elements were significantly higher at BSM compared to the polluted sites: Cd (1.25-fold of polluted sites), Ce (2.5-fold), Mn (1.1-fold), Nd (1.2-fold), Sm (1.6-fold), and V (3.2-fold). Thus, these elements are considered the main elemental contaminants by either polluted (marina, harbors) or the BSM. The threshold concentration for each element could be calculated based on two approaches: the 95th percentile and the geometric means. The 95th percentile value of the reference site was considered a threshold to expect possible toxicological impacts. The geometric mean (squared product of the concentration of the natural and polluted sites) could be another approach to identify concentration limits in clams (Table 1). These values should be considered hypothetical because they are not uniquely present in clam tissues, i.e., mixture effects are unknown. For example, if we take the most important ones above and those correlated with at least one of four physiological biomarkers (CF, GI, LT, WL), the threshold concentration of V (0.1 µg/g), La (0.7 µg/g), Ag (0.12 µg/g), Mn (3.8 µg/g), Cd (0.08 µg/g), Cr (5 µg/g) and Mo (0.67 µg/g) could serve as a flag system to identify potential sites at risk provided impacts are observed with the three morphological biomarkers (CF, GI and LT) explained below.

**Table 1.** Change in tissue levels of elements in Mya arenaria clam populations.

| Response Range | Natural Mean (5–95th) | Polluted | BSM (SLE Beluga Area of High Residency) | Threshold Concentration |
|---|---|---|---|---|
| Elements [1] | | | | |
| Ag | 0.08 (0.04–0.11) | 0.18 (0.05–0.5) [1] | 0.21 (0.06–0.7) [1] | 0.12 |
| Al | 75 (22–123) | 82 (25–171) | 127 (65–251) [2] | 102 |
| Ba | 0.6 (0.3–1) | 0.64 (0.3–1.2) | 1 (0.57–2) [2] | 0.77 |
| Be | 0.007 (0.001–0.01) | 0.007 (0.001–0.01) | 0.01 (0.004–0.02) [2] | 0.008 |
| Cd | 0.067 (0.04–0.085) | 0.08 (0.05–0.11) [1] | 0.1 (0.077–0.14) [2] | 0.08 |
| Ce | 0.69 (0.12–1.29) | 0.87 (0.25–2) | 1.7 (1–3.1) [2] | 1.08 |
| Co | 0.15 (0.05–0.26) | 0.2 (0.05–0.5) | 0.3 (0.2–0.6) [2] | 0.07 |
| Cr | 4 (0.23–8.2) | 6.3 (0.4–20) [1] | 6.7 (2–14) [1] | 5 |
| Ca | 459 (356–620) | 535 (344–973) [1] | 461 (378–580) | 495 |
| Gd | 0.05 (0.02–0.08) | 0.05 (0.02–0.14) | 0.09 (0.06–0.19) [2] | 0.07 |
| Ga | 0.04 (0.01–0.09) | 0.04 (0.01–0.14) | 0.08 (0.04–0.19) [2] | 0.06 |
| Cu | 1.8 (1.2–2.7) | 1.9 (1–3) | 2.3 (1.5–3.2) [2] | 2 |
| Fe | 214 (45–511) | 234 (52–754) | 357 (134–800) [20.67] | 276 |
| La | 0.4 (0.1–0.8) | 0.55 (0.1–1.1) | 1.1 (0.4–2) [2] | 0.7 |
| Pb | 0.06 (0.02–0.14) | 0.07 (0.02–0.14) | 0.1 (0.05–0.24) [2] | 0.08 |

**Table 1.** *Cont.*

| Response Range | Natural Mean (5–95th) | Polluted | BSM (SLE Beluga Area of High Residency) | Threshold Concentration |
|---|---|---|---|---|
| Mg | 681 (475–830) | 655 (460–834) | 500 (342–688) [2] | 583 |
| Mn | 2.4 (0.6–4.4) | 6 (1.1–44) [1] | 6.6 (2.2–17) [1] | 3.8 |
| Mo | 0.59 (0.16–0.98) | 0.95 (0.14–2.6) [1] | 0.78 (0.28–1.9) | 0.75 |
| Hg | 0.02 (0.008–0.04) | 0.02 (0.007–0.04) | 0.05 (0.03–0.08) [2] | 0.03 |
| Nd | 0.47 (0.11–0.97) | 0.78 (0.14–2.6) [1] | 0.96 (0.4–3.1) [1] | 0.67 |
| Ni | 0.33 (0.12–0.54) | 0.39 (0.14–0.9) | 0.73 (0.46–1.37) [1] | 0.49 |
| Nb | 2.5 (0.19–5.3) | 4 (0.3–12.4) [1] | 4.3 (1.3–8.9) [1] | 3 |
| P | 0.02 (0.008–0.04) | 0.02 (0.009–0.04) | 0.03 (0.01–0.06) [1] | 0.025 |
| K | 1050 (821–1384) | 978 (796–1310) [1] | 937 (820–1051) [1] | 1013 |
| Sm | 0.1 (0.03–0.19) | 0.12 (0.02–0.24) [1] | 0.2 (0.15–0.34) [2] | 0.11 |
| Sn | 0.03 (0.009–0.05) | 0.04 (0.008–0.07) | 0.05 (0.02–0.09) [2] | 0.04 |
| U | 0.03 (0.01–0.03) | 0.03 (0.01–0.05) | 0.04 (0.02–0.07) [2] | 0.04 |
| V | 0.04 (0.02–0.12) | 0.25 (0.02–0.8) [1] | 0.8 (0.05–1.4) [2] | 0.1 |
| Y | 0.03 (0.004–0.05) | 0.03 (0.005–0.05) | 0.04 (0.01–0.06) [2] | 0.035 |
| Zn | 0.16 (0.07–0.3) | 0.2 (0.07–0.51) | 0.3 (0.2–0.65) [2] | 0.22 |
| Zr | 12.8 (9.9–17.1) | 13 (9.7–16.6) | 17 (10.8–25.5) [2] | 15 |

[1] Difference from the natural site (light red) in µg/g. [2] Difference from the polluted sites ($p < 0.05$) (dark red). The green color indicates the natural variation of a clean site under no direct sources of pollution.

In the attempt to determine the biological impacts of anthropogenic activities and pollution on local clam populations at polluted sites and to assess an area of high residency of the SLE beluga whales, we determined the natural variation of the reference site (Table 2).

**Table 2.** Establishment of boundaries for natural variation of morphological changes in clams.

| Response Range | Length (mm) | Mass (g) | Age | CF | GI | Air Survival Time (Days) | Maximal Dehydration (%) |
|---|---|---|---|---|---|---|---|
| **Natural** | | | | | | | |
| Mean | 61.7 | 32.6 | 7.2 | 0.52 | 8.5 | 13.2 | 20.7 |
| Percentiles | | | | | | | |
| 25–75th | 56–66 | 23.1–41 | 6.3–8 | 0.4–0.62 | 8.2–9.5 | 11–14 | 16–24 |
| 5–95th | 42–72 | 19.4–52.1 | 5.8–9 | 0.36–0.73 | 6.8–10.6 | 10–19 | 9.8–34 |
| **Impacted (BSC and TAD)** | | | | | | | |
| Mean | 59.3 * | 26.6 * | 7.2 | 0.43 * | 8.2 * | 12 * | 24.5 * |
| Percentiles | | | | | | | |
| 25–75th | 54–63 | 16.1–31.6 | 6.1–8 | 0.3–0.52 | 7.5–9.1 | 10–13.6 | 19–29 |
| 5–95th | 49–78 | 12.3–57.1 | 5.4–10 | 0.26–0.73 | 6.3–9.1 | 7–16 | 11–34 |
| **BSM (SLE Beluga area of high residency)** | | | | | | | |
| Mean | 62.4 | 28 * | 7.3 | 0.43 * | 8.6 | 13.6 | 29.9 * |
| Percentiles | | | | | | | |
| 25–75th | 57.4–68.6 | 17.8–36.8 | 6.3–8 | 0.3–0.54 | 7.5–9.3 | 10.9–14 | 24–36 |
| 5–95th | 46–75 | 11.9–47.9 | 5.7–10 | 0.24–0.54 | 6.5–11.8 | 8–21 | 13–48 |

The mean value, 25–75th and 5–95th percentiles for over 5 years were calculated to determine the normal variation of morphological characteristics at the natural reference site MAB. Endpoints highlighted in green indicate no difference from the natural variation of the reference site; those showing significant changes from the natural variation of the reference site MAB are highlighted in light red and *. The yellow color of the BSM site indicates a mix situation i.e., intermediary state between pollution-mediated and normal variations).

This was achieved by using data from five consecutive years to determine the mean with the 5th and 95th percentiles. Because of the high number of data, we selected the

percentile approach over in the confidence interval, as it will tend to reduce as replication increases for the latter. All clams' morphological parameters (length, mass, age, CF and GI) and air survival (lethal time and maximal dehydration rate) differed between the polluted sites and the reference sites based on this approach. For example, the GI and air survival time mean values are 8.6 mm/year and 13.2 days, respectively. The natural variation ranges for GI and air-time survival are within 6.8 (5th percentile) to 10.6 mm/year (95th percentile) and 10–19 days, respectively, based on percentiles. Thus, responses occurring outside this range are considered atypical for clam populations. For example, the mean GI at polluted sites BSC and TAD is 8.2 with a 5–95th percentile range 6.3–9.1 showing lower GI responses than the natural range (6.3 to 6.8 mm/year). This suggest that clams from polluted sites display much lower growth rate. It is noteworthy that the mean GI value of the polluted sites were significantly different ($p < 0.05$) from the reference site but not with the BSM site. The BSM site generally differed from MAB for most parameters, with the exception of length, age, GI and air-time survival. In other words, clams at BSM are similar to the reference site MAB for the above endpoints, while they identical or worse to the polluted sites for mass, CF and maximal dehydration rates. Indeed, clam mass and CF were lower compared to those from the polluted and BSM sites. With respect to the maximal dehydration rate (weight loss during air emersion), clams at the polluted and BSM sites lose more weight (dehydration) than clams from the reference site MAB. Based on multiple regression analysis, age was the only explanatory predictor for dehydration rates ($r = 0.22$; $p < 0.001$), suggesting that age contributed modestly to weight loss during air emersion. Increased wet loss and decreased survival time to air emersion is a sign of pollution stress in these clam populations. We examined the yearly changes in CF, GI and air survival time. For CF, the values were closer to the 5 y mean value but a drop in CF in 2018 was observed at all sites (Figure 2). The polluted sites BSC and TAD displayed lower CF most of the times, with the exception of 2017 where no significant changes were obtained between sites. The GI was determined in clams at the reference, polluted and BSM sites for over 5 years (Figure 3). The GI at the polluted site (BSC) was lower than the reference site MAB. With respect to the other polluted site TAD, the GI was lower for the year 2019 only. Some significant changes were found for the BSM site depending on the year. The GI was reduced at BSM site in 2016 and increased at 2017. The air survival time and the dehydration rate were determined in clams over the sites at each year (Figure 4). The lethal time to air emersion was significantly increased at BSC and BSM and decreased at TAD sites in 2016. In 2017, the LT was significantly reduced at the two polluted sites TAD and BSC. The LT was reduced at BSM in 2018 but increased in 2019 and 2020. With respect to the maximum weight loss (WL), the levels were higher at the BSM site at each year compared to the reference site. The WL was also increased at the polluted site BSC in 2018, 2019 and 2020. Correlation analysis between element tissue loadings displaying significant differences from the reference sites and morphological biomarkers were performed (Table 3). The CF was significantly correlated with Ca ($r = -0.33$), Cr ($-0.41$), Mn ($r = -0.30$), Mo ($r = 0.37$), Nb ($r = -0.41$) and Nd ($r = -0.33$). Clam age was significantly correlated with Cr ($r = -0.35$), Mn ($r = -0.30$), Mo ($r = 0.43$) and Nb ($r = -0.35$). The GI was significantly correlated with Ag ($r = -0.37$) only. Air survival time was significantly correlated with Ca ($r = 0.31$), Cr ($r = 0.37$), Mn ($r = -0.44$), Nb ($r = 0.37$), Sm ($r = 0.32$) and V ($r = -0.4$). The weight loss at time of death during air emersion (dehydration rate) was significantly correlated with Ag ($r = 0.48$), Cd ($r = 0.42$), K ($r = -0.32$) and Sm ($r = 0.37$). Taken together, the strongest correlations were obtained with Ag, Cr, Mn and V with either one of the three morphological biomarkers (CF, GI and LT) including the WL during air emersion. Multiple regression analysis revealed (results not shown) that the CF was best predicted by V ($\beta = 0.11$), Sm ($\beta = -0.57$) and Mn ($\beta = -0.1$), showing that Sm and Mn reduced clam CF. The GI was best predicted by Mn ($\beta = 0.06$), Sm ($\beta = 4.4$), Ni ($\beta = 0.66$), Ag ($\beta = -4.75$) and V ($\beta = -0.06$), indicating that Ag and V had a detrimental effect on growth. The LT was significantly predicted by tissue levels of Mn ($\beta = 0.23$), Sm ($\beta = 14.8$) and V ($\beta = -4$). The maximal WL during air emersion was significantly predicted by Cd ($\beta = 153$),

Cr ($\beta = -1$) and Nd ($\beta = 6.4$). The following elements were the most often observed across morphological biomarkers: Ca (three out of five morphological indices), Cr (three out of five indices), Mn (three out of five indices) and Nb (three out of five indices). Among these, Cr and Mn had the strongest correlations.

**Table 3.** Correlation analysis of tissue elements and biomarker data. Only elements showing significant difference with the reference site are shown. Significant correlations (($r \geq 0.3$, $p \leq 0.05$) are in **bold**).

|  | LT | WL | Ag | CF | GI | Ca | Ce | Gd | Ga | Fe | La | Mn |
|---|---|---|---|---|---|---|---|---|---|---|---|---|
| LT | 1 | | | | | | | | | | | |
| WL | −0.15 | 1 | | | | | | | | | | |
| Age | **−0.49** | 0.25 | 1 | | | | | | | | | |
| CF | **−0.34** | −0.01 | **0.6** | 1 | | | | | | | | |
| GI | **0.31** | −0.19 | **−0.72** | −0.04 | 1 | | | | | | | |
| Ca | **0.31** | −0.17 | −0.29 | **−0.33** | 0.13 | 1 | | | | | | |
| Ce | 0.14 | **0.43** | 0.01 | −0.22 | −0.01 | 0.22 | 1 | | | | | |
| Gd | **0.48** | 0.22 | **−0.3** | **−0.4** | 0.22 | **0.47** | **0.85** | 1 | | | | |
| Ga | **0.39** | 0.16 | −0.29 | **−0.31** | 0.26 | **0.45** | **0.79** | **0.91** | 1 | | | |
| Fe | **0.52** | −0.03 | **−0.43** | **−0.39** | **0.31** | **0.57** | **0.5** | **0.79** | **0.9** | 1 | | |
| La | 0.12 | **0.47** | −0.01 | −0.18 | 0.04 | 0.09 | **0.95** | **0.76** | **0.68** | **0.34** | 1 | |
| Mn | **0.34** | 0.02 | **−0.33** | **−0.3** | 0.29 | **0.56** | **0.38** | **0.53** | **0.53** | **0.53** | **0.34** | 1 |
| K | 0.15 | **−0.32** | −0.16 | −0.13 | 0.04 | 0.27 | −0.17 | −0.01 | 0.04 | 0.25 | **−0.31** | 0.21 |
| Sm | **0.32** | **0.37** | −0.16 | −0.29 | 0.15 | 0.24 | **0.92** | **0.89** | **0.75** | **0.49** | **0.95** | **0.45** |
| Ag | −0.11 | **0.48** | 0.21 | −0.25 | **−0.37** | 0.06 | **0.37** | 0.28 | 0.09 | 0.02 | **0.36** | 0.06 |
| V | **−0.4** | 0.16 | 0.29 | 0.29 | −0.04 | 0.08 | **0.56** | **0.32** | **0.38** | 0.11 | **0.54** | 0.17 |
| Ni | 0.19 | **0.41** | −0.06 | −0.22 | 0.09 | 0.28 | **0.97** | **0.88** | **0.88** | **0.63** | **0.92** | **0.43** |
| Nd | 0.23 | 0.09 | −0.23 | **−0.33** | 0.16 | **0.71** | **0.45** | **0.54** | **0.63** | **0.64** | **0.3** | **0.71** |
| Mo | **−0.44** | 0.14 | **0.43** | **0.37** | −0.19 | −0.27 | 0.03 | −0.13 | −0.06 | −0.16 | 0 | −0.14 |
| Pb | −0.02 | 0.21 | 0.03 | −0.02 | 0.04 | **0.45** | **0.49** | **0.43** | **0.48** | **0.38** | **0.39** | **0.44** |
| Cd | 0.11 | **0.42** | 0.01 | −0.2 | 0.01 | **0.33** | **0.72** | **0.62** | **0.42** | 0.23 | **0.71** | **0.48** |
| Mg | 0.03 | **−0.36** | −0.11 | 0.06 | 0.02 | 0.05 | **−0.59** | **−0.51** | **−0.38** | −0.17 | **−0.6** | −0.25 |
| Nb | **0.37** | −0.08 | −0.35 | **−0.41** | 0.22 | **0.77** | 0.26 | **0.49** | **0.42** | **0.51** | 0.15 | **0.74** |
| Cr | **0.39** | −0.09 | −0.34 | **−0.42** | 0.18 | **0.78** | 0.25 | **0.5** | **0.43** | **0.53** | 0.12 | **0.76** |
| Hg | 0.11 | 0.5 | 0.03 | −0.12 | 0.03 | −0.01 | **0.88** | **0.74** | **0.59** | 0.25 | **0.97** | **0.45** |
| Sm | −0.22 | 1 | | | | | | | | | | |
| Ag | −0.01 | 0.36 | 1 | | | | | | | | | |
| V | −0.1 | 0.43 | 0.13 | 1 | | | | | | | | |
| Ni | −0.16 | **0.9** | 0.29 | **0.54** | 1 | | | | | | | |
| Nd | 0.26 | **0.36** | 0.11 | 0.18 | **0.54** | 1 | | | | | | |
| Mo | 0.01 | −0.11 | 0.05 | **0.47** | 0.04 | −0.07 | 1 | | | | | |
| Pb | 0.17 | **0.38** | 0.02 | **0.53** | **0.52** | **0.61** | **0.31** | 1 | | | | |
| Cd | 0.04 | **0.73** | 0.45 | 0.35 | **0.65** | **0.5** | 0.1 | **0.53** | 1 | | | |
| Mg | −0.04 | **−0.6** | −0.37 | **−0.48** | **−0.56** | −0.21 | **−0.58** | **−0.53** | **−0.69** | 1 | | |
| Nb | **0.31** | **0.32** | 0.13 | 0.02 | **0.3** | **0.87** | −0.17 | **0.47** | **0.52** | −0.19 | 1 | |
| Cr | **0.32** | **0.31** | 0.13 | 0.02 | **0.3** | **0.88** | −0.18 | **0.45** | **0.54** | −0.21 | **0.9** | 1 |
| Hg | −0.38 | **0.94** | **0.35** | **0.54** | **0.84** | 0.21 | −0.01 | **0.33** | **0.71** | −0.61 | 0.12 | 0.11 |

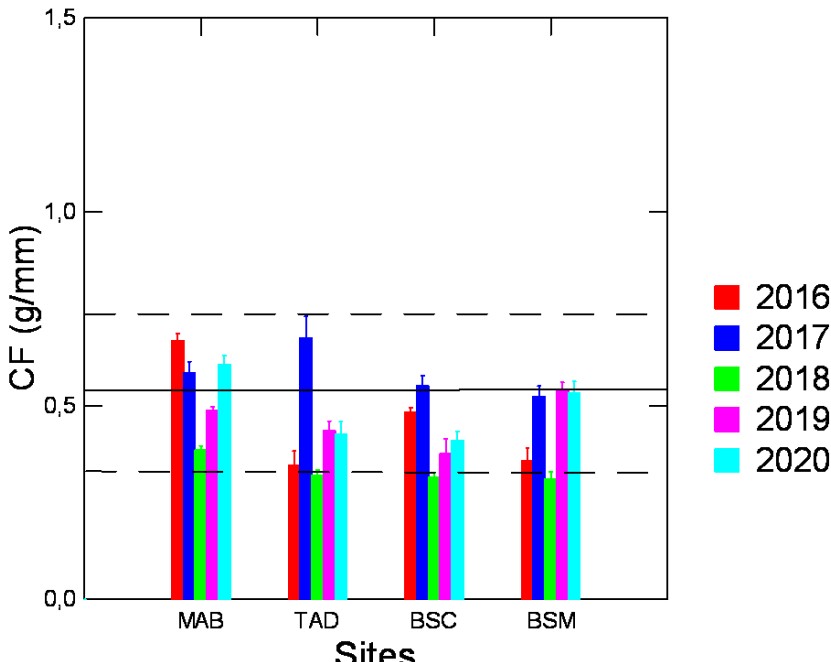

**Figure 2.** Yearly changes in condition factor in *Mya arenaria* clams by site. The CF (total weight/shell length) was determined in clams at each site for over the 5-year period. The filled and doted lines represent the mean and the 5–95th percentile of the reference site MAB to depict the natural variation of CF in clams.

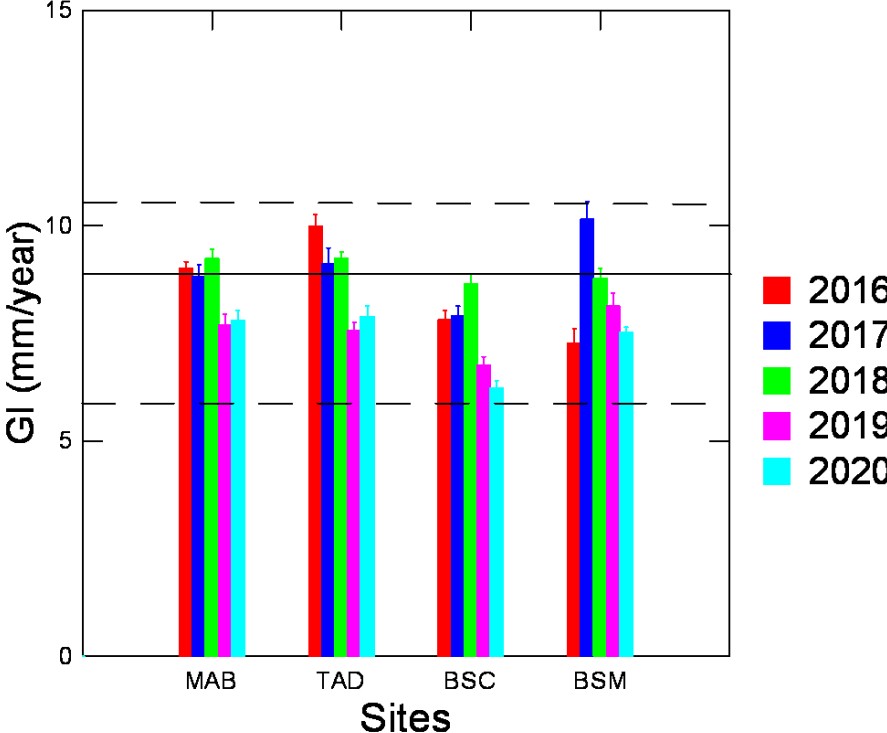

**Figure 3.** Temporal changes in the growth index of clams. The GI (shell length/age) was determined in clams at each site for over the 5-year period. Legends as in Figure 2.

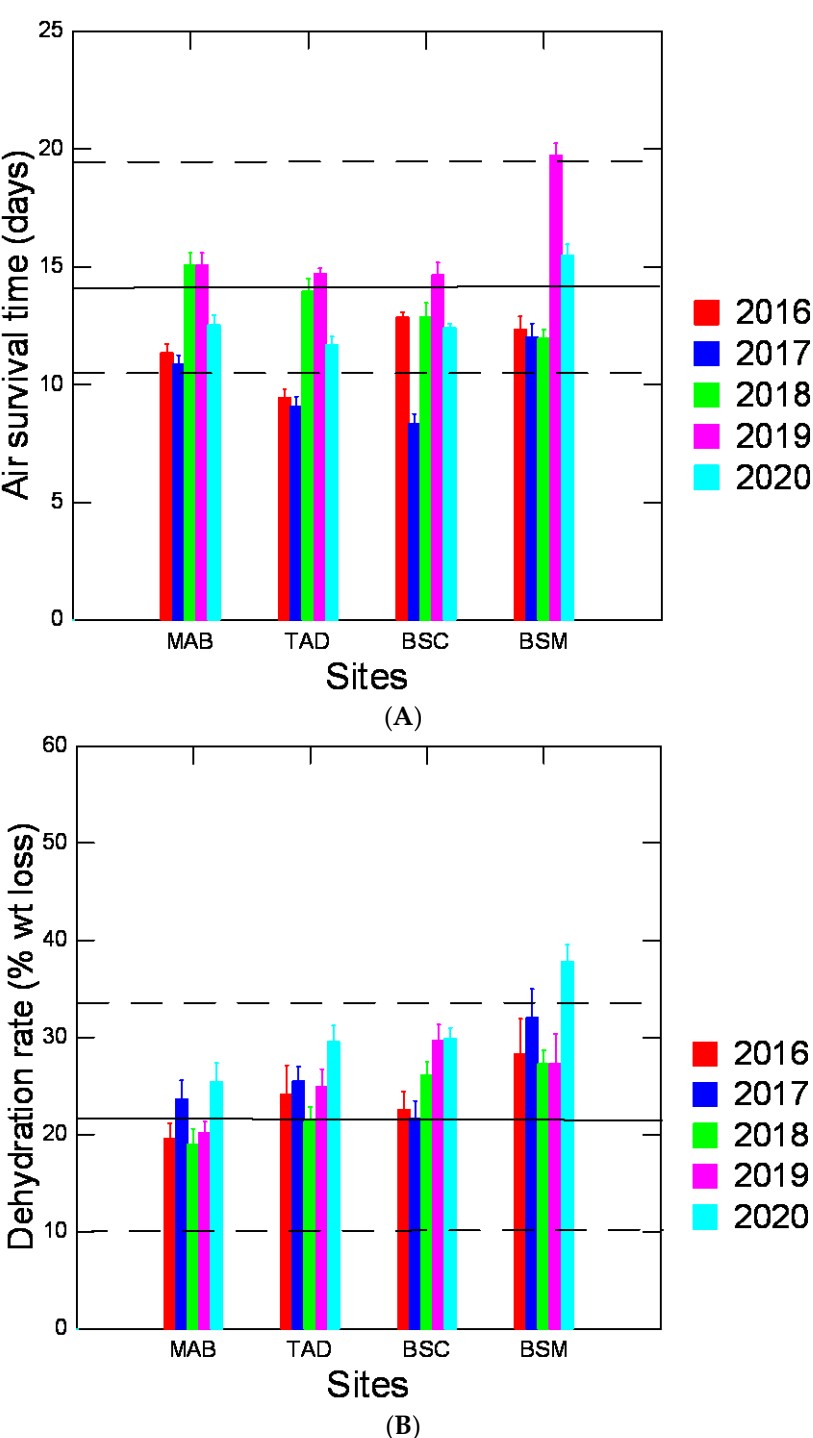

**Figure 4.** Temporal change in the air survival time and dehydration rate in clams. Clams were collected at low tide in the morning and processed immediately for air survival (**A**) and weight loss (dehydration rate) (**B**). Legends as in Figure 2.

In an attempt to seek out long-term trends, Fourier transformation and rescaled range analyses were preformed (Figure 5). The rescale range analysis of the CF was H = 0.941, 0.977 and 0.964, suggesting that these changes were not random (white noise, H = 0.5) and show a long-term trend toward a decrease in the CF over time. Fourier transformation was used to filter out reduced inter-individual variability for each year. With respect to CF (Figure 5A), the reference site changes revealed a strong drop in 2018 and returned levels to those found in 2016 in 2020, hence a period of 5 years. At the polluted sites, the amplitudes

changes were stronger and occurred within a period of 2 years compared to the reference site. For example, the maximum amplitude changes at the reference site were 0.7/0.35 = 2 compared to 0.69/0.28 = 2.5 for the polluted stress and 0.55/0.33 = 1.7 at BSM. Based on this analysis, BSM shows more similarity with the reference site. The GI changes over the 5-year period at the reference site, and it was fairly stable, with amplitudes oscillating between 9.1 and 7.2 (1.26) over the years with no clear period pattern. This is consistent with an H = 0.69 showing some downward trend in time. At the polluted sites, the amplitudes were more important with a maximal change of 10/5.8 = 1.7, and the long-term decrease is stronger with an H value of 0.901. At BSM, the amplitudes were also much higher with a maximum change of 10.5/6.3 = 1.7 similar to the polluted sites. The H index at 0.991 suggests a stronger downward trend than both the reference and polluted sites. With respect to LT, an increase in the amplitudes occurred in 2018 and 2017 at the reference site with no clear periodic pattern. The rescale range analysis (Hurst exponent) was 1.015, 1.0815 and 0.615 for the reference, polluted and BSM, respectively, suggesting that the LT increased more at the polluted sites, while this trend weakened with increased variability in the LT at BSM.

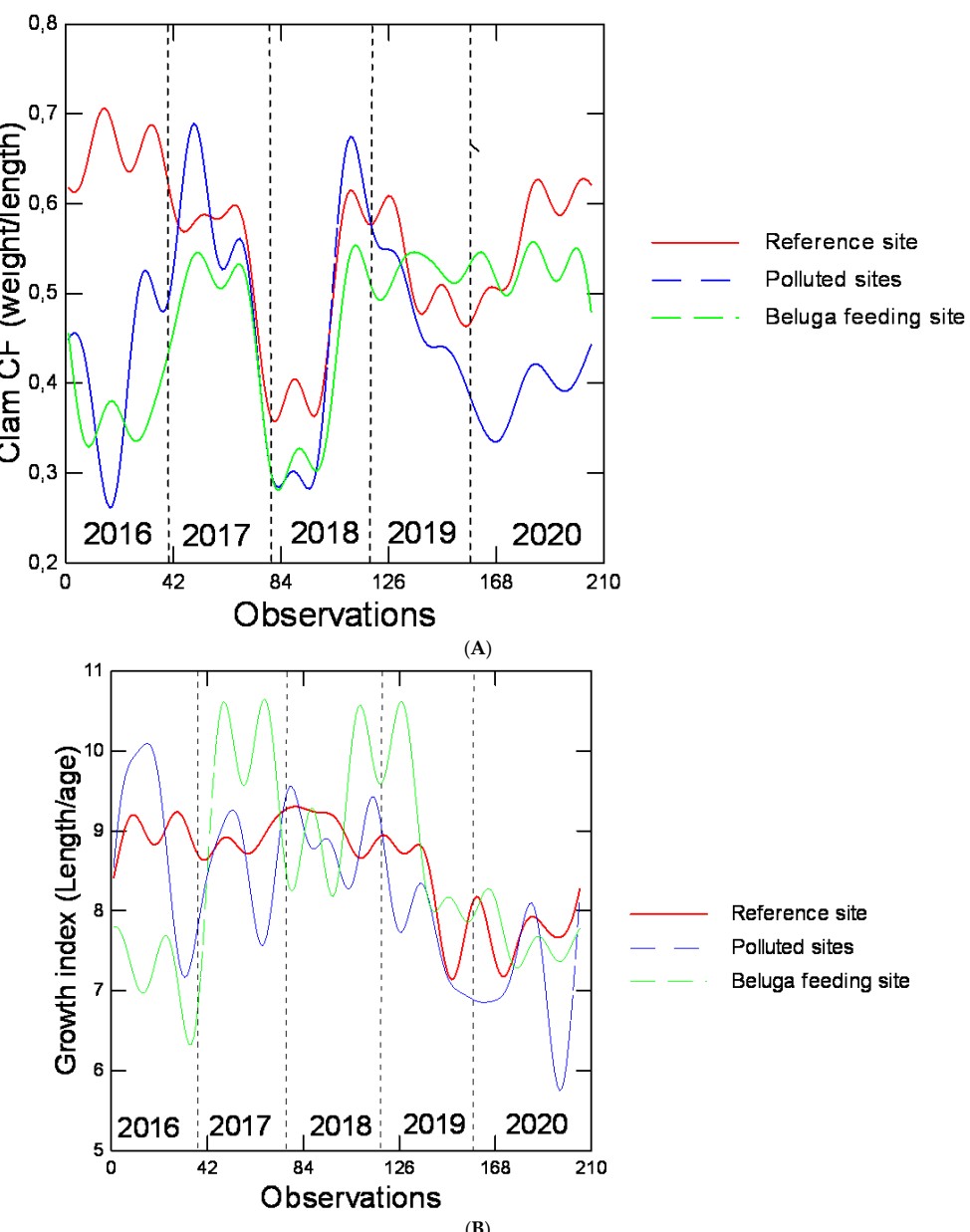

**Figure 5.** *Cont.*

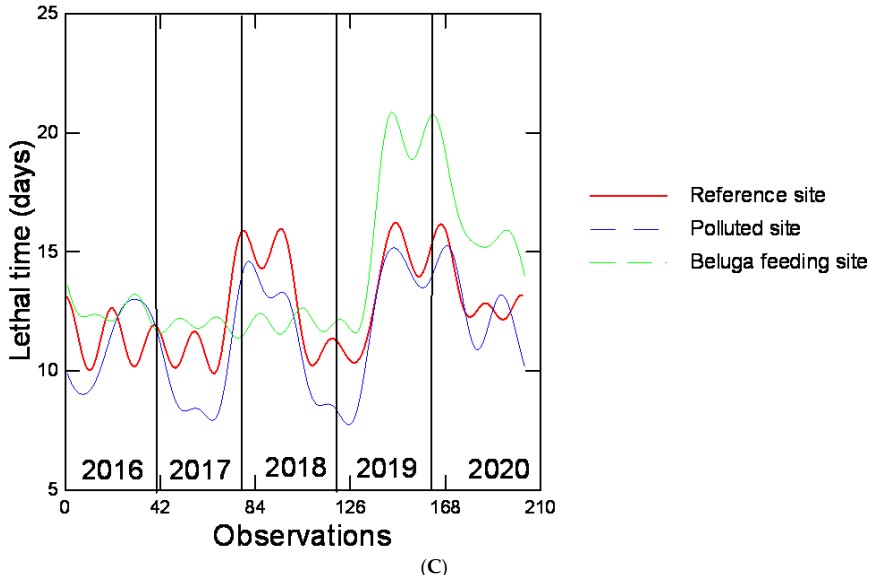

(C)

**Figure 5.** Trend analysis of biomarker data. The 5 years of biomarker data were filtered by Fourier analysis (12 principal frequencies) to observe the global trends in time for CF (**A**), growth index (**B**) and lethal time (**C**). Each year (40 observations) of collection is separated by dotted lines. The reference site (MAB), polluted sites (BSC and TAD) and SLE beluga area (BSM) are marked in different colors.

## 4. Discussion

Elemental analysis of clam tissues revealed that the following elements were readily ($\geq$2-fold) increased in clam tissues over the time survey at the polluted and BSM sites compared to the reference site MAB: V, Mn, Ce, Hg, Ag, Ni, Sm and Ga. The absence of a positive correlation between these eight elements with K suggests that these elements behaved as particles. However, K (as with sodium, Ca and Mg) is physiologically regulated and could have contributed, in part at least, to the lack of correlation. Despite this, K tissue levels were significantly lower in clams at the polluted and BSM sites, suggesting a decreased capacity to retain K in tissues. Loss of K could indicate electrolyte loss, perhaps by reduced respiration (metabolic activity) and Na/K-pump activity [16]. In a previous study with snails and mussels, exposure to mercury decreased K levels in tissues at both lethal and sublethal concentrations, providing some insights into the toxic effects of this contaminant in invertebrates [17]. It is noteworthy that nitrogen elimination ($NH4^+$) follows $Na^+/K^+$-ATPase activity in membrane pores in gills [18], suggesting that low $K^+$ in tissues could be associated to altered nitrogen turnover. The highest increase in elements in clam tissues was V with a 20-fold increase at BS compared to the reference site. This element is a natural contaminant of crude oil, which is transported by commercial ships in the Saguenay Fjord or unknown sources of crude oil are inadvertently released in this area. This suggests that clams at the polluted and BSM sites are exposed to crude oil or diesel fuel contamination. Indeed, high levels of V were found in ashes from heavy fuel oil (diesel) combustion in marine engines [19]. Although V from fuel combustion would be V oxides, $V^{2+}$ was shown to induce oxidative stress and metallothionein levels in *Mytilus* sp. tissues [20]. This is consistent with a previous study showing increased levels of metallothioneins in the digestive gland of clams located near a wharf compared to outside the harbor area [21]. This study also revealed that V in tissues (0.43–0.91 µg/g) were in the same range as those found in the present survey and negatively impacted the CF based on correlation analysis. Although the CF was not significantly correlated with V in tissues in the present study, V levels were negatively correlated with the air survival time of clams, which was in turn related to CF, age and the GI. It is interesting that the increase in V tissues in the 2006 study was 2.4-fold compared to 6.3-fold at the same sites in the present study; hence, V contamination in clams has been increasing over the last decades. In clams,

*Ruditapes philippinarum,* chronically exposed to Cd, a threshold tissue concentration of 7 µg/g was associated to Cd-induced toxicity [22]. This concentration was about 50 higher than the 95th percentile value of Cd concentration in tissues in this study. This suggests that Cd on its own is unlikely to be a major player of the observed effects. With respect to V, the levels of V in tissues at roughly 0.1 µg/g was related to metallothionein levels in blue mussels, suggesting effects at the biochemical level [20]. Tissue mean concentrations of V in clams was circa 2.5 and 20 times higher than the reference site, suggesting that V could have contributed to toxicity based on the physiological health indicators CF, GI and LT. The most bioaccumulated elements in clams were Mn, Ce, La and Sm, mainly found as fuel additives in the form of methylcyclopentadienyl manganese tricarbonyl [23], cerium oxides and oil-treatment soaps to improve the performance of low octane petroleum distillates or reduce particulate emissions (Sm and La). La is also relatively abundant in the earth's crust and is used in fuel additives to reduce particulate emissions and also in rechargeable batteries for electric/hybrid cars [24]. This is consistent with the significant correlations of tissue Ce levels with V and Sm. Mn was recognized as a neurotoxicant manifesting as Parkinson-like symptoms [25]. Studies on the toxicity of the Mn organic complex found in gasoline in bivalves and fish are scarce. Exposure of the freshwater bivalve *Anodontites trapesialis* to ionic Mn (0.5 mg/L) suppressed gill metabolic rates [26,27]. This is in keeping with the negative correlation with the clam mass and CF found in the present study. However, the reduced metabolism (lower production of reactive oxygen species during respiration) could have contributed to increase air survival time (significant correlation between Mn and LT) in clams. Fuel additives comprise other elements in addition to Mn, such as Nd (increased at polluted sites), Sm (increased at polluted sites) and Ce (increased at BSM). Ce levels in clams were significantly increased 2.5-fold at BSM only. The third most accumulated metal was Ag. Although no direct source of Ag is found in this area, Ag contamination is a result of human activity, such as smelting and coal burning activity, erosion of natural sources, mining, and municipal wastes [28]. Hence, the elemental profiles suggest contamination principally from combustion emissions of fuel and crude oil release, which is consistent with harbors and sustained boat traffic. It is noteworthy that contaminants were sometimes higher at BSM compared to the two polluted sites MAB and TAD, suggesting that this food source could pose a risk to organisms feeding on this resource.

This study also permitted us to examine the temporal changes of clam health by pollution based on the proposed morphological biomarkers (CF, GI and LT) [10]. Based on these endpoints, long-term memory analysis revealed time effects deviating from random changes in time (white noise) at both polluted and BSM sites. A decrease in the CF and GI with an increasing trend in the LT over time was globally observed, i.e., all sites confounded. However, the decrease in the CF was stronger at the polluted sites and showing more intense amplitude changes (instability). At BSM, the CF tended to increase in contrast to the reference and polluted sites. With respect to GI, the decrease in time was much more severe at the polluted sites and growth instability in (high amplitude changes), while no long-term changes were observed at BSM. The LT tended to increase more steeply in time at the polluted and BSM compared to the reference site. It is possible that decreased metabolism from the elements discussed above could have contributed to the increased LT. However, the increased air survival time occurred at the expense of growth and size/condition (mass/length ratio). Reduced growth and condition of freshwater mussels was found in those caged downstream of a municipal effluent [29]. This was further supported by the fact that mussel recruitment was sparse downstream of the effluent discharge. In another study, shell length of the wild mussel *Elliptio dilatata* was smaller with increasing tissue concentrations of Cd, Pb and Zn from a river supporting zinc mining activities [30]. The BSM site contained clams that were mostly contaminated by Ce, Co, Ga, La, Hg, Ni, Sm and V. This raises the issue of beluga whales being contaminated by these elements. In beluga tissues collected in the Arctic near Alaska, some elements were present in beluga skin samples [31]. For example, these metals/elements were higher at some sites compared to Cook Inlet site: Ag (6.6-fold), Cd (4.4-fold), V (4.6-fold) and Hg

(9.6-fold). This supports the hypothesis that SLE beluga could be contaminated as well, perhaps in part through feeding on local clam populations. It appears that Hg levels in beluga in the western Artic was not solely related to diet or Hg emissions but followed decanal oscillations [32]. Indeed, Hg contamination seems to follow large oscillations in surface water temperatures spanning many decades, the so-called Pacific or Atlantic Decadal Oscillations. The clams and perhaps beluga contamination profile could show a cyclic period from the Atlantic decadal oscillations. These oscillations occur at time scales covering many decades (10–80 years), much larger than the present time scale of the present survey. Although outside the scope of this survey, this could explain why the GI and LT did not change much over the years.

In marine mussels acclimated to low salinity, the air survival time was lower in the summer (warming) months compared to colder temperatures closer to the sites in the present study [33]. Interestingly the decrease in LT was attenuated by adaptation to salinity stress, suggesting that long-term adaptation to environmental stress would increase mussel resiliency to air exposure. The CF was also reduced to mussels that acclimated to salinity reductions. It is possible that decreased metabolism (producing less oxidative stress) could have contributed to increased air survival time. A negative relationship between air survival and lipid peroxidation in mitochondria was observed in a recent study investigating species susceptibility to environmental stressors using physiologically based biomarkers [12]. The increased air survival was favored with increased mass and shell length but with decreased oxidative metabolism. Given that some of the elements were found to decrease oxidative metabolism in bivalves, this could have contributed to air survival in addition to decrease growth and condition factor. A recent meta-analysis on freshwater taxa revealed that nitrate levels, a general chemical signature of polluted environments, were generally associated to reduced growth and developmental abnormalities for various taxa principally composed of 91% vertebrates and 9% invertebrates [34]. Based on the present study, it appears that the decrease in growth also occurred in bivalves with a concomitant increase in air survival time. Oyster populations in an estuary were shown to have reduced CF at the urban-impacted sites [35]. In this study, the levels of vitellogenin-like proteins were also lower, suggesting a decrease in energy transfer in embryo and could contribute to reduced recruitment over time. Decreased vitellogenin-like proteins and CF were previously observed at one of the polluted sites (BSC) in clams [36].

## 5. Conclusions

This study revealed that clams were more contaminated with V, Mn, La, Ce and Ag, which are mainly related to the combustion of fuel (Mn, Ce, La), crude oil contamination (V) and urban pollution (Ag). The strong association of La with Sm suggests its use as an oil additive. The contamination profile of the BSM sites differed from the polluted sites, especially for Ga, La, Hg, Ni and Sm. The clams at the polluted sites displayed lower CF and GI and were more vulnerable to air emersion stress compared to the reference sites. At the BSM, only the CF and dehydration rate were significantly decreased. Long-term analysis revealed that resistance to air emersion tended to increase over time at the expense of low condition and growth indices. Based on the data presented, the clams from the Baie Sainte-Marguerite, an area of high residency of the SLE beluga, where feeding is occurring, are contaminated with respect to the reference site, which can pose a health risk to this emblematic and endangered whale species.

**Author Contributions:** C.A. performed the biomarker analyses, data analysis and manuscript preparation. S.T. and N.M. provided support for field sampling logistics and manuscript preparation. F.G. is the principal investigator and participated in data statistical analyses and manuscript preparation. All authors have read and agreed to the published version of the manuscript.

**Funding:** This research was funded by Parks Canada and Environment and Climate Change Canada.

**Data Availability Statement:** The data are available upon demand by request to the corresponding author.

**Acknowledgments:** The authors wish to thank the numerous field technicians from Parks Canada at the Saguenay–St. Lawrence Marine Park for field clam collection over the years, particularly Laurence Lévesque and Sarah Duquette for their assistance in the planning and logistics of field sampling and Camille Bégin-Marchand for mapping the area. The authors also wish to thank the SÉPAQ for their assistance in the field sampling at the BSM site. This project was jointly supported by Parks Canada and Environment and Climate Change Canada.

**Conflicts of Interest:** The authors declare no conflict of interest.

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
