# Peer review of "Spatio-Temporal Variation of Elemental Contamination and Health of Mya arenaria Clam in the Saguenay–St. Lawrence Marine Park"

_applsci, doi:10.3390/app12031106_

Round 1

Reviewer 1 Report

The authors noted that the content of toxic elements was different depending on the sampling area with significant changes over time. Thus, the authors observed a high amount of toxic elements (Ag, Mn, Ce, Ga, La, Hg, etc.) in the body of animals collected from polluted sites. Also, factor condition, growth index, and air survival time were decreased in animals collected from polluted areas in contrast to a reference unpolluted site.

Mya arenaria is a filter-feeding organism that can bioaccumulate a number of pollutants and these can affect its physiological state. At the same time, the physiological parameters of clams might be influenced by the physicochemical traits of the environment in which they live, especially those related to nutrition.

For this reason, the authors should present more data on the living environment from which the organisms were collected and analyzed. From this point of view, a description of the food sources in the sampling places should be made. Are the food sources similar in the sampling areas? What are the qualitative and quantitative differences between food sources? Other factors that may affect the physiological state of the collected specimens, such as temperature, oxygen concentration, etc should also be considered, in order to finally determine the factor with the major influence on the development and physiological state of the studied organisms. The authors should also provide data on the concentration of the toxic elements studied in the water and sediment column in order to have a better picture of all the factors involved. Could these differences in toxic elements alone explain the differences observed in the development of organisms?

After a more detailed analysis of the living conditions, on the one hand, and of the concentration of pollutants in the sampling area, on the other hand, the major factor responsible for the physiological differences of the analyzed organisms could be established more clearly.

The article itself is well written, but in order to be published, it needs a series of additional data as suggested above (respectively, a more detailed description of the physicochemical and trophic conditions of the sampling sites).

Author Response

Answers to comments are provided in blue here and in the revised document.

The authors noted that the content of toxic elements was different depending on the sampling area with significant changes over time. Thus, the authors observed a high amount of toxic elements (Ag, Mn, Ce, Ga, La, Hg, etc.) in the body of animals collected from polluted sites. Also, factor condition, growth index, and air survival time were decreased in animals collected from polluted areas in contrast to a reference unpolluted site.

Mya arenaria is a filter-feeding organism that can bioaccumulate a number of pollutants and these can affect its physiological state. At the same time, the physiological parameters of clams might be influenced by the physicochemical traits of the environment in which they live, especially those related to nutrition.

For this reason, the authors should present more data on the living environment from which the organisms were collected and analyzed. From this point of view, a description of the food sources in the sampling places should be made. Are the food sources similar in the sampling areas? What are the qualitative and quantitative differences between food sources? Other factors that may affect the physiological state of the collected specimens, such as temperature, oxygen concentration, etc should also be considered, in order to finally determine the factor with the major influence on the development and physiological state of the studied organisms. The authors should also provide data on the concentration of the toxic elements studied in the water and sediment column in order to have a better picture of all the factors involved. Could these differences in toxic elements alone explain the differences observed in the development of organisms? These sites were extensively examined over the past 20 years, which clearly showed contamination problems as shown in various references provided in the text, e.g.:

  1. Blaise C, Gagné F, Gillis PL, Eullaffroy P. Polychaetes as bioindicators of water quality in the Saguenay Fjord (Quebec, Canada): a preliminary investigation. J Xenobiotics 3: 1-3, 2013.
  2. Blaise C, Gagné F, Burgeot T. Three simple biomarkers useful in conducting water quality assessments with bivalve mollusks. Environ Sci Pollut Res. 24: 27662-27669,
  3. Fortin G, Pelletier M. Synthèse des connaissances sur les aspects physiques et chimiques de l'eau et des sédiments du Saguenay: zones d'intervention prioritaire 22 et 23. Technical report. Montréal: Environment Canada, Environmental Conservation, St. Lawrence Centre; 1995.
  4. Gagné F, Blaise C, Pellerin J, Pelletier E, Strand Health status of Mya arenaria bivalves collected from contaminated sites in Canada (Saguenay Fjord) and Denmark (Odense Fjord) during their reproductive period. Ecotoxicol. Environ. Saf. 64: 348-361, 2006.
  5. Gagné F, Blaise C, Pellerin J, Pelletier E, Douville M, Gauthier-Clerc M, Viglino L. Sex alteration in soft-shell clams (Mya arenaria) in an intertidal zone of the Saint Lawrence River (Quebec, Canada). Comp Biochem Physiol C 134, 189–198, 2003.
  6. Gagnon F, Tremblay T, Rouette J, Cartier JF. Chemical risks associated with consumption of shellfish harvested on the north shore of the St. Lawrence Rivers lower estuary. Environ. Health Perspect. 112: 883–888, 2004.

.

After a more detailed analysis of the living conditions, on the one hand, and of the concentration of pollutants in the sampling area, on the other hand, the major factor responsible for the physiological differences of the analyzed organisms could be established more clearly. We added this information on basic properties at each sites at lines 114-17 and 171-177. However, we did not perform the complete elemental analysis of pore water sediments and sediment solids (because of budget restriction)..

The article itself is well written, but in order to be published, it needs a series of additional data as suggested above (respectively, a more detailed description of the physicochemical and trophic conditions of the sampling sites). This is provided in the revised manuscript.

Thank you for the comments.

Reviewer 2 Report

A very interesting article that analyzes very important bioindicators. Did bioethical licensing be required to conduct research at calm population? If so, please mention it in the method. I also recommend that you emphasize the need to conduct such research and the possibility of its practical application. Please explain why the position inhabited by whales was chosen. Why is it so important (introduction chapter) It is also advisable to separate the conclusions chapter. It is a summary of the results of your work; should give new knowledge in science. I am aware that not all typing errors were caught while reading. Therefore, I am asking for their exact correction. 

Author Response

Answers to comments are provided in blue here and in the revised document.

A very interesting article that analyzes very important bioindicators. Did bioethical licensing be required to conduct research at calm population? No this is not required for invertebrates.If so, please mention it in the method.

 I also recommend that you emphasize the need to conduct such research and the possibility of its practical application. This study is an application of the 3 simple biomarker approach that was used in clam populations from polluted sites (as confirmed by elemental analysis in tissues). See lines78-80.

Please explain why the position inhabited by whales was chosen. Why is it so important (introduction chapter). Done see lines 86-88.

It is also advisable to separate the conclusions chapter.  Done see line 429.

It is a summary of the results of your work; should give new knowledge in science. I am aware that not all typing errors were caught while reading. Therefore, I am asking for their exact correction. Corrections were done in the revised manuscript.

Round 2

Reviewer 1 Report

I read the revised manuscript and noted that the authors made the necessary changes, improving the quality of the manuscript. Therefore, the article may be published in Applied Sciences in revised form.